 **eLIFE**

# Transgenerational inheritance of ethanol preference is caused by maternal NPF repression

Julianna Bozler, Balint Z Kacsoh, Giovanni Bosco*

Department of Molecular and Systems Biology, Geisel School of Medicine at Dartmouth, Hanover, United States

**Abstract** Rapid or even anticipatory adaptation to environmental conditions can provide a decisive fitness advantage to an organism. The memory of recurring conditions could also benefit future generations; however, neuronally-encoded behavior isn't thought to be inherited across generations. We tested the possibility that environmentally triggered modifications could allow 'memory' of parental experiences to be inherited. In *Drosophila melanogaster*, exposure to predatory wasps leads to inheritance of a predisposition for ethanol-rich food for five generations. Inhibition of Neuropeptide-F (NPF) activates germline caspases required for transgenerational ethanol preference. Further, inheritance of low NPF expression in specific regions of $F_1$ brains is required for the transmission of this food preference: a maternally derived *NPF* locus is necessary for this phenomenon, implicating a maternal epigenetic mechanism of NPF-repression. Given the conserved signaling functions of NPF and its mammalian NPY homolog in drug and alcohol disorders, these observations raise the intriguing possibility of NPY-related transgenerational effects in humans.

DOI: https://doi.org/10.7554/eLife.45391.001

## Introduction

To what extent is personality and behavior predetermined at birth? Philosophers and scientists alike have struggled with this question, and many have settled on the *tabula rasa*, or *blank slate* perspective. This long-standing notion posits we are without form or direction until our individual experiences shape us. Over the past decades, however, evidence has accumulated that suggests parental environment can have significant phenotypic consequences on the next generation, thus eroding this notion of a blank slate. The Dutch Hunger Winter Study was one of the first documented examples of ancestral experiences influencing subsequent generations. Children conceived in the Netherlands during the World War II blockade, and ensuing famine, had higher rates of obesity and diabetes (*Heijmans et al., 2008*; *Schulz, 2010*; *Stein et al., 1975*). More recent studies have found that neurological and mental health conditions also appear to have persistent impact on the next generations (*Yeshurun and Hannan, 2019*). Further, risk factors for children of Holocaust survivors, such as reduced cortisol sensitivity, have been linked to methylation state of the glucocorticoid receptor promoter, and increased methylation in offspring has been associated with paternal diagnosis of posttraumatic stress disorder (*Yehuda et al., 2014*).

Studied largely in the public health context, there are limited examples of environmental inheritance that can be experimentally tested. Genetic model systems are indispensable for understanding molecular mechanisms of causation. For example, male mice trained to associate fear with an odor, transmitted sensitivity of this odor to their sons. In this instance, researchers concluded that offspring possessed an increased abundance of sensory neurons specific to the same odor their fathers were trained to fear (*Dias and Ressler, 2014*). Similarly, environmental enrichment activities can

**\*For correspondence:**
Giovanni.Bosco@dartmouth.edu

**Competing interests:** The authors declare that no competing interests exist.

ameliorate behavioral defects of mutant mice defective in long-term potentiation and memory. This behavioral rescue is heritable to the next generation through the activation of an otherwise latent p38 signaling cascade (*Arai et al., 2009*). Parental exposure to toxins and nutritional challenges also can change germline information, affecting growth and metabolism of future generations (*Carone et al., 2010*; *Chen et al., 2016*; *Sharma et al., 2016*; *Skinner et al., 2013*). These few examples suggest that parental environment can have a profound impact on subsequent generations. Elucidating mechanisms behind these environmentally triggered epigenetic programs is essential for a complete understanding of the foundational principles upon which biological inheritance is based.

*Drosophila melanogaster* females, when cohabitated with endoparasitoid wasps, shift to prefer ethanol food as an egglaying substrate, where ethanol food protects Drosophila larvae from wasp infection (*Kacsoh et al., 2013*). *Drosophila suzukii* similarly shifts egglaying preference to food with atropine, giving its progeny protection against wasps (*Poyet et al., 2017*). Ethanol preference in *D. melanogaster* is linked to a decrease in Neuropeptide F (NPF) in the female brain (*Kacsoh et al., 2013*), consistent with previous work on NPF (*Shohat-Ophir et al., 2012*), and its mammalian homolog NPY that has been studied in the context of drug addiction (*Gonçalves et al., 2016*; *Landayan and Wolf, 2015*). NPY modulation governs ethanol consumption in rats (*Thiele et al., 1998*) and is implicated in human alcohol abuse disorders (*Mayfield et al., 2002*; *Mottagui-Tabar et al., 2005*). This behavioral output is believed to be a consequence of the NPF/NPY role in the rewards pathway, with NPF signaling being rewarding (*Desai et al., 2013*; *Shao et al., 2017*). NPF activity is considered representative of the motivational state of the fly (*Krashes et al., 2009*; *Landayan and Wolf, 2015*). Several recent studies also have shown that 'stressful' experiences regulate NPY/NPF levels, providing a link between environmental cues and NPF/NPY signaling (*Broqua et al., 1995*; *Sah et al., 2009*; *Shohat-Ophir et al., 2012*). Here we present findings that link maternal environmental conditions to inheritance of an altered reward pathway *via* depressed NPF signaling and a preference for ethanol.

## Results

### Inheritance of ethanol preference

Drosophila were cohabitated with female wasps for four days, then separated. Flies were then placed into embryo collection chambers for 24 hr. Embryos were divided into two cohorts and each developed in the absence of adult flies or wasps. One cohort was used to propagate the next generation and never exposed to ethanol food; the second cohort was used in the ethanol preference assay and then discarded (*Figure 1a*).

Wasp-exposed Canton-S flies lay approximately 94% of their eggs on ethanol food (*Figure 1b*). This behavior persists in their offspring despite the $F_1$ generation never having direct interaction with wasps (*Figure 1b*). Ethanol preference in $F_1$ was less potent, with 73% of the eggs laid on ethanol food (p=8.6e$^{-7}$, *Supplementary file 1*). Remarkably, this inherited ethanol preference persisted for five generations, gradually reverting back to the mock exposed baseline (*Figure 1b*). These observations were replicated in an additional wild type Oregon R (OR) strain (*Supplementary file 2*), suggesting that the phenomenon is not specific to a particular genetic aberration or background. This indicates that inheritance of ethanol preference is not a permanent germline change, but rather it is a reversible trait. Ethanol preference was measured for two days for initial experiments (*Figures 1–3* and *Figure 1—figure supplement 1*); day one and day two showed similar trends, suggesting that flies do not habituate to ethanol, nor does the preference fade over the course of the experiment (*Supplementary file 3*).

To explore the required neural signaling to the germline, mutants defective for long-term memory were assayed. Previous studies have shown that flies defective in long-term memory exhibit an ethanol preference only in the presence of wasps, but not after wasp removal. The long-term memory mutant $Orb2^{\Delta Q}$ produced offspring with an ethanol preference when the embryo collection for the $F_1$ generation was conducted in the presence of wasps, but this ethanol preference was greatly reduced in offspring collected post-wasp exposure (*Figure 1c*). Similar results were found for the memory mutant $amn^1$ (*Figure 1—figure supplement 2a*). To further probe the role of long-term memory, a conditional knockdown of $Orb2$ was employed: In this system, upon ingestion of the

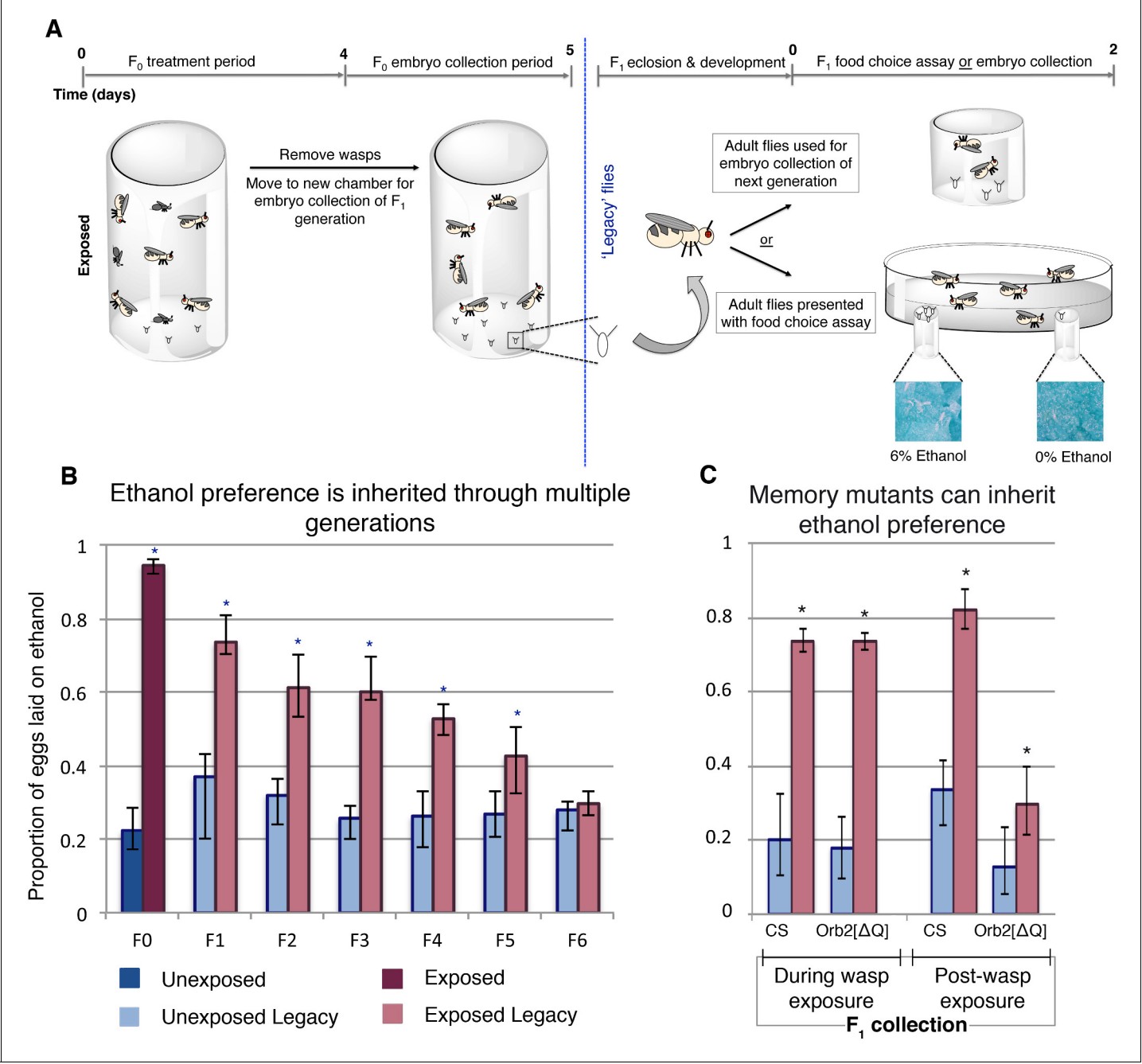

**Figure 1.** Maternally inherited ethanol preference persists for multiple generations. Schematic of experimental flow is shown (**A**). Flies are exposed to wasps for a period of four days prior to egg collection. The descendants from either wasp-exposed or unexposed treatment groups, termed 'legacy' flies, are separated from the previous generation and reared until maturity. Legacy flies are either used to propagate the next generation, or are assayed for ethanol preference. Flies from a particular generation are referred to as $F_n$, where n denotes the number of generations removed from the treatment. For example, the treatment group itself is $F_0$, whereas their direct offspring are $F_1$. Ethanol preference is quantified as proportion of eggs laid on ethanol food (**B**), illustrating that this behavior is heritable through the $F_5$ generation. Flies with deficient long-term memory (Orb2[ΔQ]) were tested for transgenerational inheritance of ethanol along side the wild type control strain (CS) (**C**). Embryos ($F_1$ legacy flies) were collected during wasp exposure or in the 24 hr period following the wasp exposure; both CS and Orb2[ΔQ] exposed legacy flies are able to inherit the ethanol preference. Asterisk indicates p-value of <0.05 from a Mann-Whitney U test. Error bars are bootstrap 95% confidence intervals. Color-coding of bar charts indicates treatment and generation; dark blue (unexposed), light blue (unexposed legacy), dark magenta (exposed), light magenta (exposed legacy).
DOI: https://doi.org/10.7554/eLife.45391.002

The following figure supplements are available for figure 1:

**Figure supplement 1.** Temporal dynamics of wasp exposure effect inheritance of ethanol preference.

*Figure 1 continued on next page*

*Figure 1 continued*

DOI: https://doi.org/10.7554/eLife.45391.004

**Figure supplement 2.** Intact long-term memory is dispensable for the transmission of ethanol preference.

DOI: https://doi.org/10.7554/eLife.45391.003

**Figure supplement 3.** $F_1$ ethanol preference has distinct characteristics from those of the parental $F_0$ generation (pertaining to *Figure 1*).

DOI: https://doi.org/10.7554/eLife.45391.005

**Figure supplement 4.** Global transcriptional changes in the female head.

DOI: https://doi.org/10.7554/eLife.45391.006

drug RU486, a RNA hairpin to *Orb2* is expressed in the mushroom body, a region of the brain essential for long-term memory (*Bozler et al., 2017*; *Roman et al., 2001*). These experiments provide the advantage of generating offspring with wild type long-term memory and avoid the possible developmental complications of mutant lines. Legacy flies collected during the wasp exposure and subsequently tested did exhibit an ethanol preference (*Figure 1—figure supplement 1b*). However, legacy flies from the parental RU486 treatment collected post-wasp exposure did not display an ethanol preference. These data provide insight in two ways: First, functional long-term memory is not a compulsory requirement to generate ethanol-preferring offspring. Second, the mutant data suggests that intact long-term memory is not required to inherit ethanol preference. Given that ethanol preference in the absence of wasps is long-term memory dependent, this experiment reveals that the neuronal signaling is different for maintained ethanol preference in the $F_0$ and $F_1$ flies (*Bozler et al., 2017*).

We confirmed previous findings, where, following a wasp exposure, $F_0$ flies have an ethanol preference that persists for more than a week, returning to baseline after ten days (*Figure 1—figure supplement 1a*). Sister cohorts of $F_1$ flies were collected at two time points along this $F_0$ ethanol preference decay; one immediately following wasp exposure (brood 1), and a second, ten days post wasp exposure (brood 2). Brood two did not display an inherited ethanol preference, suggesting that wasp exposure does not inflict a permanent change in the $F_0$ germline (*Figure 1—figure supplement 1d*). Again, these findings replicated in OR flies (*Supplementary file 2*), indicating that these observations are robust and not dependent on the context of a particular genetic background.

To explore further the role of time and dynamics of wasp exposure, multiple generations of flies were exposed to wasps. We found that inherited ethanol preference can be enhanced with successive generations of wasp exposure (*Figure 1—figure supplement 1e*). This trend did not repeat when nonconsecutive generations were repeatedly exposed to wasps (*Figure 1—figure supplement 1f*). This suggests that the enhancing effect observed in the successive exposures is time sensitive and may be linked to the ethanol preference of the parental flies.

Several other factors point to distinctions between the $F_0$ and $F_1$ ethanol preference behavior. Male $F_1$ legacy flies, mated to naïve females produced offspring ($F_2$) with an ethanol preference (*Figure 1—figure supplement 3a*). Additionally, 14–16 day old $F_1$ flies displayed an ethanol preference, demonstrating that $F_1$ flies do not have an ethanol preference decay curve similar to that of the $F_0$ (*Figure 1—figure supplement 3b*).

## Transcriptional changes

Global transcriptional changes in the female head across generations were examined with RNA sequencing. Heads from the exposed legacy $F_1$ and $F_2$ generation were collected and normalized to their respective unexposed legacy group. These results were then compared with the $F_0$ generation, which was previously reported (*Bozler et al., 2017*). Analysis of the $F_0$ data detected 98 differentially expressed transcripts (15 down and 83 up) (*Figure 1—figure supplement 4*, *Supplementary file 4*). $F_1$ and $F_2$ heads showed very few differentially expressed transcripts, 4 and 5 transcripts respectively. Of the differentially expressed transcripts, no transcript was shared between groups. These data indicate that although wasp exposure itself results in global transcriptional changes in the female head, this observation does not hold true for the subsequent generations.

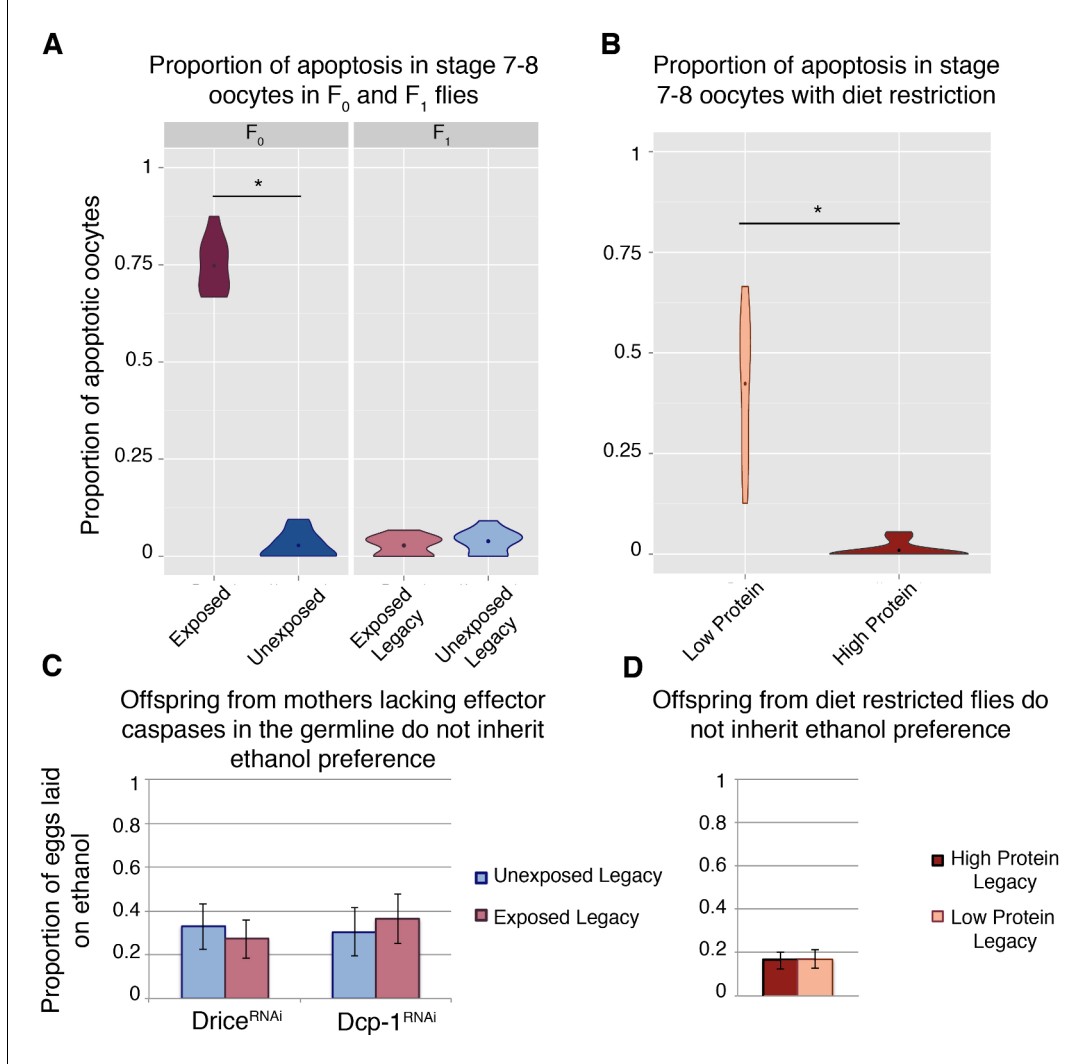

**Figure 2.** Germline apoptosis and activated caspases play a role in the inheritance of ethanol preference. Apoptosis in stage 7–8 egg chambers was quantified in $F_0$ and $F_1$ (legacy) flies (**A**); wasp exposure leads to elevated levels of apoptosis but is not persistent across the next ($F_1$) generation. Similarly, apoptosis was quantified in stage 7–8 egg chambers in flies fed different diets: Flies fed a protein-restricted diet have elevated levels of stage 7–8 oocyte apoptosis (**B**). Genetic knockdown of the germline effector caspases, Drice or Dcp-1, was achieved by expressing a RNA hairpin in the female germline (driven by the maternal-αtubulin-Gal4). Offspring from these flies were collected and tested for ethanol preference: Ethanol preference is not inherited from mothers with Dcp-1 or Drice knockdown (**C**). Offspring from the diet treatments were similarly tested; progeny from protein-restricted parents don't inherit an ethanol preference (**D**). Points within violin plots denote the group mean. Sample size was 10 for each experimental group. Asterisk indicates a p-value of <0.05 from a Mann-Whitney U test. Error bars are bootstrap 95% confidence intervals.
DOI: https://doi.org/10.7554/eLife.45391.007

## Germline caspases are necessary

Mid-oogenesis germline apoptosis (stage 7–8 oocytes) is triggered upon wasp exposure (*Figure 2a*) (*Kacsoh et al., 2018b*; *Kacsoh et al., 2018a*; *Kacsoh et al., 2015*). However, this wasp response is not heritable like the ethanol preference behavior, and $F_1$ females do not exhibit germline apoptosis (*Figure 2a*). Nevertheless, maternal germline knockdown of the effector caspases *Dcp-1* and *drice* (maternal-αtubulin-Gal4 > UAS-Dcp-1[RNAi], and maternal-αtubulin-Gal4 > UAS-Drice[RNAi] respectively) produce offspring without an ethanol preference, regardless of parental treatment (*Figure 2c*). Although protein-starvation triggers germline apoptosis similar to wasp exposure (*Figure 2b*), offspring from mothers with starvation-induced apoptosis do not inherit an ethanol preference (*Figure 2d*). This indicates that germline apoptosis in and of itself is not sufficient for inheritance of ethanol preference.

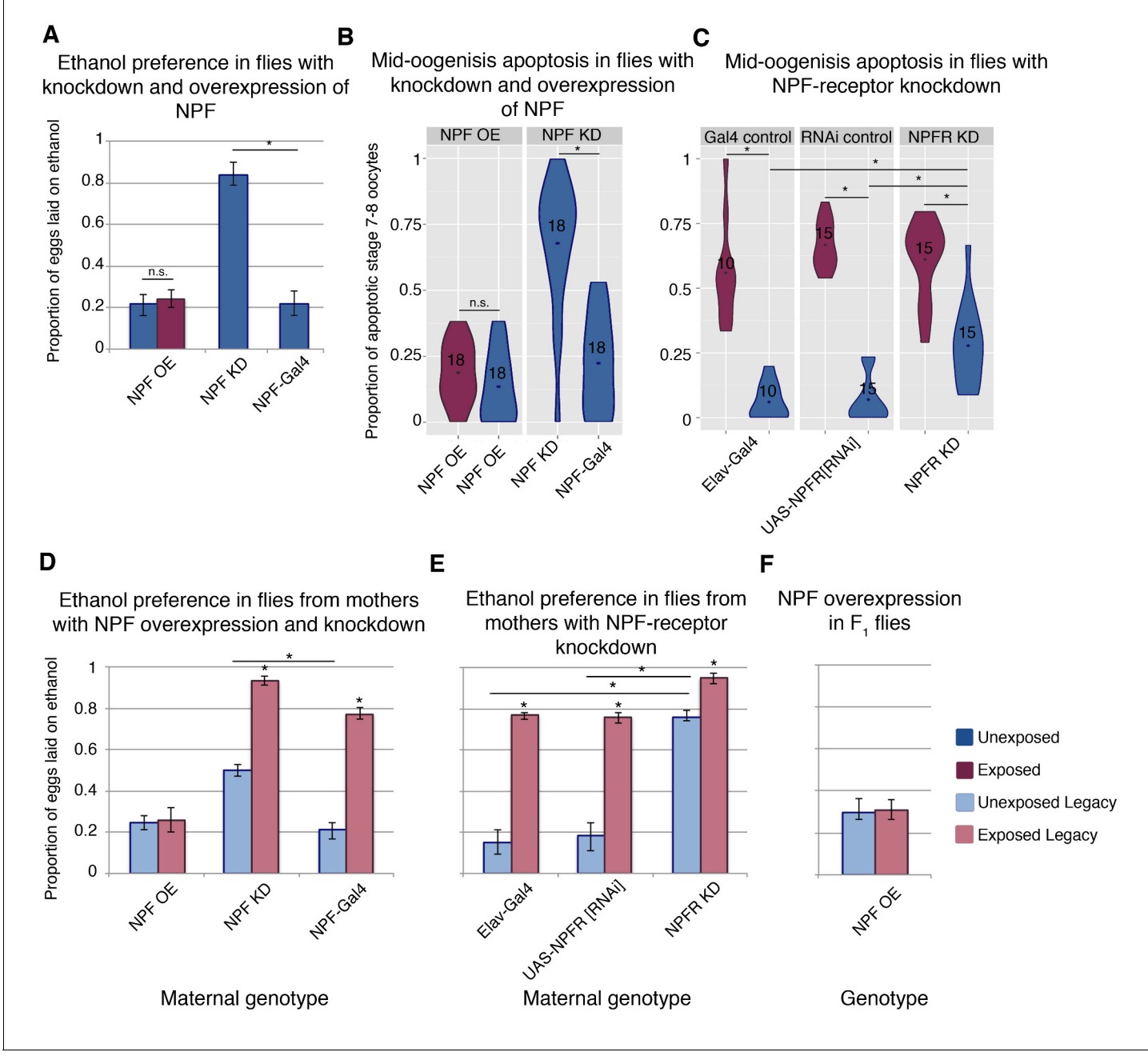

**Figure 3.** NPF affects ethanol preference and germline apoptosis. Genetic manipulation of NPF levels can alter ethanol preference (A). NPF overexpression (OE) or knockdown (KD) was achieved in the NPF-expression pattern (NPF-Gal4). This genetic manipulation of NPF can alter levels of germline apoptosis as well (B). Knockdown of the NPF-receptor in neurons (using the pan neuronal driver Elav-Gal4) leads to increased germline apoptosis (C). $F_1$ legacy flies have altered ethanol preference depending on the maternal NPF genotype (D). Similarly, maternal knockdown of the NPF-receptor in neurons can drive inheritance of ethanol preference in legacy flies (E). Genetic manipulation of NPF in the $F_1$ legacy flies (overexpression in the NPF-expression pattern) can alter the inheritance of ethanol preference (F). Points within violin plots denote the group mean; the number within the violin plot indicates sample size for the group. Asterisk indicates a p-value of <0.05 from a Mann-Whitney U test. Error bars are bootstrap 95% confidence intervals. Color-coding of charts indicates treatment and generation; dark blue (unexposed), light blue (unexposed legacy), dark magenta (exposed), light magenta (exposed legacy).

DOI: https://doi.org/10.7554/eLife.45391.008

The following figure supplement is available for figure 3:

**Figure supplement 1.** Ethanol preference in $F_1$ legacy flies for transgene control lines.
DOI: https://doi.org/10.7554/eLife.45391.009

## NPF and its receptor modulate germline apoptosis

NPF is known to play a role in food seeking, ethanol consumption, and numerous other reward pathways, and NPF levels decrease in the fan shaped body of female fly brains following wasp exposure (*Kacsoh et al., 2013*). Even in the presence of wasps, overexpression of NPF (NPF-Gal4 >UAS NPF) inhibits ethanol preference, while in the absence of wasps, knockdown of NPF (NPF-Gal4 >UAS-NPF [RNAi]) is sufficient to induce the ethanol preference behavior (*Figure 3a*). Given this NPF modulation of ethanol preference in females, we asked whether NPF also signaled to germline cells, triggering caspases and apoptosis. Strikingly, NPF knockdown induces mid-oogenesis apoptosis in the absence of wasps (*Figure 3b*), while overexpression of NPF results in no elevation in germline apoptosis even in the presence of wasps (*Figure 3b*). Similarly, NPF-receptor (NPFR) knockdown in neurons (Elav-Gal4 >UAS-NPFR[RNAi]) alone leads to significantly elevated levels of apoptosis (28%, when compared to parental line controls p=$6.2e^{-4}$ and $1.5e^{-4}$), and this effect is enhanced with wasp exposure (61%, p=$7.0e^{-4}$) (*Figure 3c*). Taken together these observations link ethanol preference behavior and mid-oogenesis apoptosis in the $F_0$ females, where both processes are likely caused by changes in NPF and NPFR signaling.

## Changes in NPF trigger transgenerational inheritance of ethanol preference

We speculated that the NPF-triggered changes in $F_0$ behavior and germline might also correlate with observed changes in offspring. For these experiments, it was critical to ensure that $F_1$ flies did not share the maternal genotype, and a crossing scheme was devised to avoid progeny with transgene expression (see Materials and method section). Legacy flies from mothers with NPF knockdown exhibit ethanol preference, even in the absence of wasp exposure (*Figure 3d* and *Figure 3—figure supplement 1*). Inherited ethanol preference is enhanced when the parental NPF knockdown flies are exposed to wasps (*Figure 3d*). By contrast, NPF overexpression in $F_0$ mothers exposed to wasps produced offspring lacking the ethanol preference (*Figure 3d*). NPFR knockdown experiments mirror these findings: Maternal NPFR knockdown produces offspring with an ethanol preference compared to unexposed control lines; again this effect is enhanced when NPFR knockdown is paired with wasp exposure (*Figure 3e*). Interestingly, overexpression of NPF in $F_1$ flies blocks ethanol preference in the exposed $F_1$ legacy group (*Figure 3f*), raising the possibility that $F_1$ legacy flies inherit NPF in a repressed or low expression state.

We therefore hypothesized that regulation or depression of NPF might be a means of this behavioral inheritance. Global changes in NPF RNA were not detected in either the $F_0$ or $F_1$ female heads (*Figure 4—figure supplement 1*). However, antibody staining allowed for a region specific examination of NPF protein levels (*Figure 4a* and (*Figure 4—figure supplement 2*). Anti-NPF signal has clear overlap with the NPF-Gal4 driving the cd8-GFP reporter (*Figure 4a*). The fan shaped body has previously been implicated in ethanol preference, and therefore, was a focus in this experiment (*Kacsoh et al., 2013*). NPF protein levels measured through immunofluorescence were significantly reduced in the fan shaped body of $F_0$, $F_1$, and $F_2$ (two-generations exposed) flies (*Figure 4b*). We note that NPF was not reduced in all regions of the $F_1$ and $F_2$ brains, as intensity of P1 neurons was not reduced in either the $F_1$ or $F_2$ flies, although significant reduction was observed in P1 neurons of $F_0$ flies (*Figure 4c*).

Given the observed link between depressed NPF and oocyte apoptosis, it is notable that $F_1$ flies do not have germline apoptosis. It is possible that apoptosis is due to a localized decrease in NPF not shared between the two generations; perhaps the apoptosis is triggered by other NPF neurons or synapses. It is also conceivable that other neural processes are altered in the flies that we did not detect, decoupling the apoptosis and ethanol preference behaviors in the later generations.

## Maternal chromosomal inheritance of ethanol preference behavior

To determine whether maternal or paternal exposure were equally important for transgenerational inheritance of ethanol preference, wasp-exposure and mating were controlled in two separate experiments. First, mated females were exposed to wasps in the absence of male flies. Second, wasp-exposed males were mated to naïve virgin females, removing the maternal exposure as a factor. Interestingly, $F_1$ offspring from an exclusively maternal wasp exposure inherit ethanol preference, while $F_1$ offspring from an exclusively paternal exposure did not (*Figure 5a*). Importantly, this

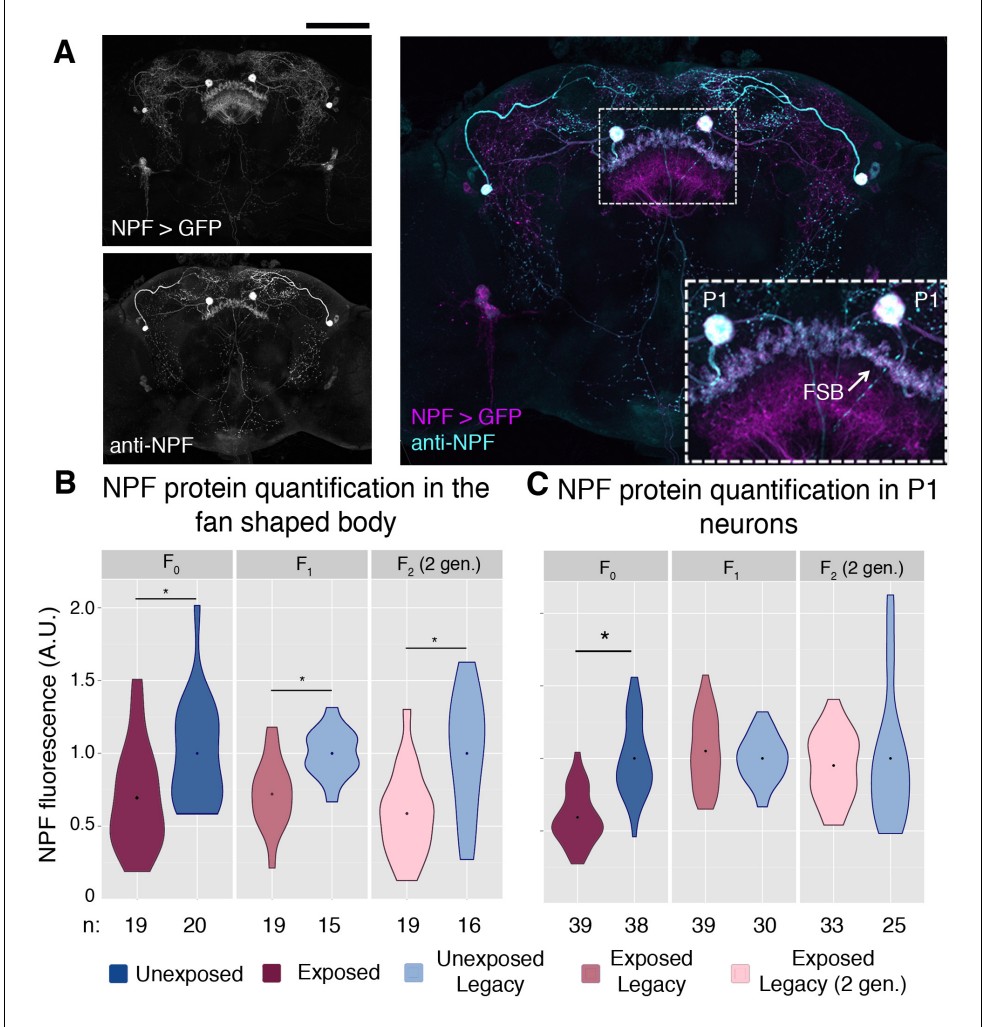

**Figure 4.** NPF protein is reduced in the fan shaped body following wasp exposure. NPF antibody staining has a similar pattern to that of NPF-Gal4 expression in an adult female brain, inset shows a magnification of the two large P1 neurons and the fan shaped body (FSB) (A). NPF protein levels are reduced in the fan shaped body across generations (B). NPF depression in P1 neurons is observed only in the $F_0$ generation (C). Points within violin plots denote the group mean. Sample size (n) is indicated at the bottom of the graph for each group. Asterisk indicates a p-value of <0.05 from a Mann-Whitney U test. Scale bar is 100 microns.

DOI: https://doi.org/10.7554/eLife.45391.010

The following figure supplements are available for figure 4:

**Figure supplement 1.** mRNA quantification of NPF in female fly heads.

DOI: https://doi.org/10.7554/eLife.45391.011

**Figure supplement 2.** Region of interest for NPF protein quantification.

DOI: https://doi.org/10.7554/eLife.45391.012

demonstrates a key difference between the $F_0$ males and the $F_1$ males, and it may point to an 'activation phase' and 'maintenance phase' of the epigenetic program. We have previously reported that female flies require sight to induce a behavioral response to wasp exposure (*Kacsoh et al., 2015*). In further support of the maternal contribution to the inheritance of ethanol preference, blind female flies (ninaB[1]; UAS-ninaB) mated to CS males did not produce offspring with an ethanol preference (*Figure 5b*). In the reciprocal experiment, blind fathers mated to CS females did generate ethanol-preferring offspring following a wasp exposure (*Figure 5b*).

The maternal epigenetic inheritance of ethanol preference could be conferred by chromosomal elements and/or cytoplasmic factors. If ethanol preference is inherited through a chromatin mark, then chromosome parental-origin tests should reveal a requirement for maternal chromosomal

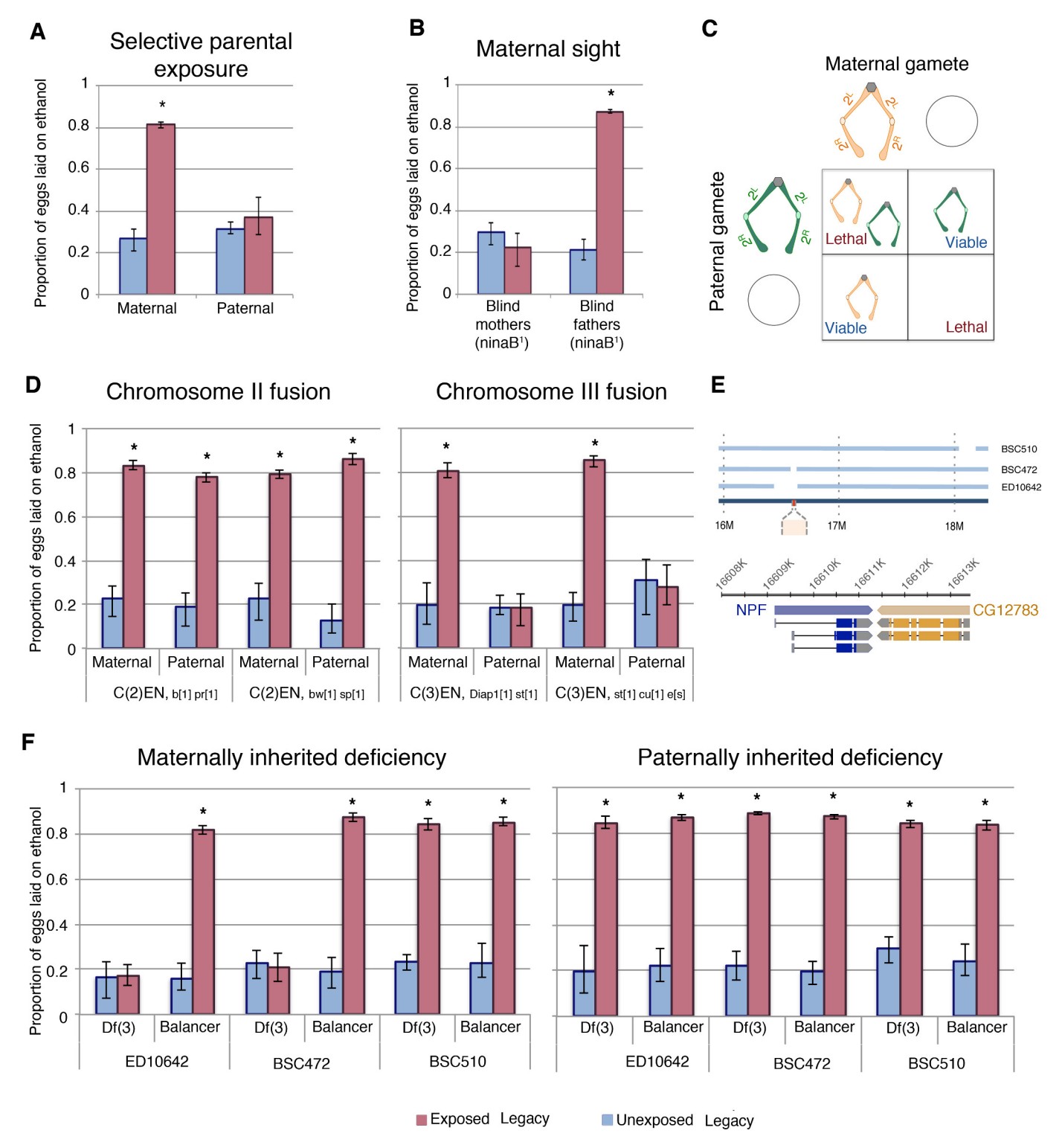

**Figure 5.** Maternal chromosome three is required for inherited ethanol preference. Experiments with exclusively maternal or paternal wasp exposure demonstrate that maternal wasp exposure is necessary for ethanol preference inheritance (**A**). 'Blind' flies, with a mutation in *ninaB* were used to test the requirement of sight: maternal or paternal ninaB[1] flies (mated with wild type counterparts) were wasp exposed and offspring tested for ethanol preference (**B**). Schematic of compound chromosome 2; progeny inherit both copies of the chromosome from either maternal or paternal source and are identified based on chromosomal markers (**C**). Flies receiving either maternal or paternal copies of the compound chromosome two are able to inherit the ethanol preference, but compound chromosome three must be maternally derived to facilitate inheritance of ethanol preference (**D**). The
*Figure 5 continued on next page*

*Figure 5 continued*

relative location of NPF (red) on chromosome three and the deleted region of the deficiency stock is shown in a diagram (**E**). The inheritance of ethanol preference was observed in flies receiving an intact maternal NPF locus on a balancer chromosome and not in flies from receiving a maternal NPF deficiency (Df3) chromosome: Paternal inheritance of the NPF deficiency had no effect on transmission of ethanol preference (**F**). Asterisk indicates a p-value of <0.05 from a Mann-Whitney U test. Error bars are bootstrap 95% confidence intervals.

DOI: https://doi.org/10.7554/eLife.45391.013

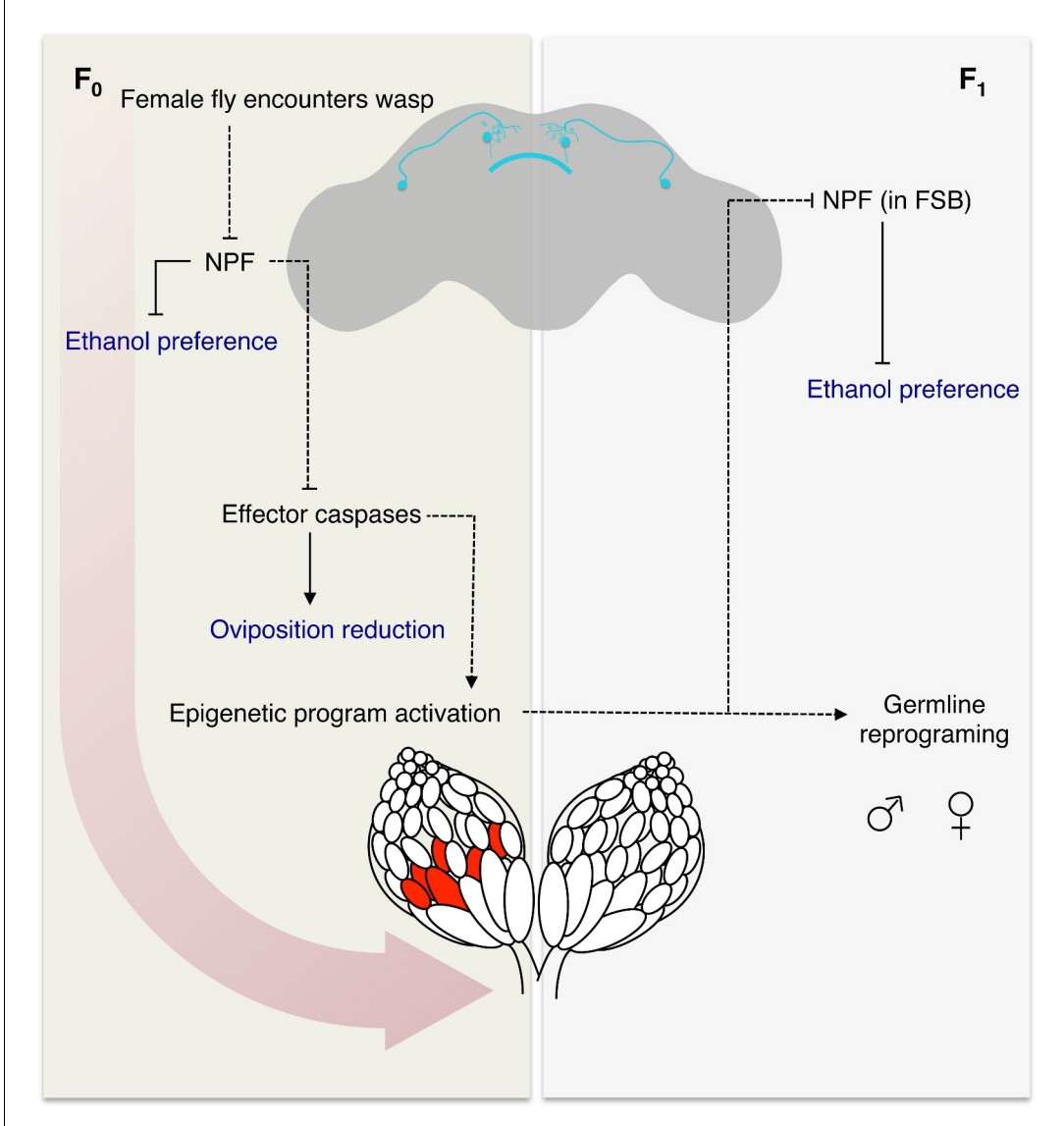

**Figure 6.** Model for fly-wasp mediated ethanol preference. A female fly encounters a wasp, based in part on visual signals, leading to a cascade of physiological and behavioral changes. One of the initiating factors following wasp exposure is the depression of NPF in the female fly brain. Under normal conditions, NPF inhibits the ethanol preference behavior and caspase mediated germline apoptosis. Therefore, the reduction of NPF triggers ethanol preference and germline caspases. Activation of the germline effector caspases Dcp-1 and Drice in turn reduced egg laying and participates in the epigenetic reprograming of the female germline and chromosome 3. The epigenetic program is passed to both sexes of the $F_1$ generation, in that both male and female progeny can pass on the ethanol preference. Further, legacy $F_1$ female flies inherit depressed NPF in the fan shaped body (FSB), which drives the ethanol preference behavior. Model legend: Measured behavioral outputs are in blue, the dashed lines indicate a speculative or unknown mechanism of action.

DOI: https://doi.org/10.7554/eLife.45391.014

inheritance. However, if inheritance is conferred through a maternally contributed cytoplasmic factor, then passage of all chromosomes through paternal gametes should have no effect, as wasp-exposed females can still maternally deposit molecules and organelles into the oocyte. To test what maternal components may be conferring inheritance of ethanol preference we first focused on chromosomal elements using attached, or compound, chromosomes. Flies where each of the two homologs are fused cannot make haplo-chromosome gametes. Instead, they can only make gametes with one or zero copies of the fused chromosome, and therefore $F_1$ flies inherit 'pairs' of homologs that are entirely maternally or paternally derived (*Figure 5c*). In this manner, we tested each of the two major autosomes for parent-of-origin effects. Using phenotypic markers, flies were sorted as having either a maternal or paternal exclusive homolog pair and assayed for ethanol preference. Chromosome-II fusion flies had similar results when inheriting exclusively maternal or paternal Chromosome-II elements (*Figure 5c*). Chromosome-III fusion flies also had inheritance of ethanol preference when receiving both copies of Chromosome-III maternally. However, flies with both copies of Chromosome-III from their fathers failed to inherit an ethanol preference (*Figure 5d*). This observation has at least three implications: Most importantly, this indicates that some element on Chromosome-III must be inherited from wasp-exposed mothers in order for ethanol preference behavior to be passed on to $F_1$ legacy flies. This also suggests that maternal copies of the Chromosome-X, Chromosome-II or cytoplasmic factors, if important, are not sufficient for inheritance of ethanol preference. Lastly, that oocytes giving rise to eggs with zero copies of maternal Chromosome-II still confer ethanol preference, indicates that exclusion of maternal chromosomes itself does not generally interfere with transgenerational inheritance.

## A maternal NPF locus is required for epigenetic inheritance

To further delineate what parts of maternally derived Chromosome-III were required for transgenerational inheritance of ethanol preference we tested chromosomes with well defined deletions. As NPF has previously been shown to control ethanol preference behavior, we speculated that the NPF locus on Chromosome-III may be a target of maternal epigenetic reprogramming (*Shohat-Ophir et al., 2012*). We also observed that $F_1$ legacy flies inherit low levels of NPF expression specifically in the fan shaped body of the brain (*Figure 4a–b*), consistent with the possibility that $F_1$ flies inherit repressed NPF expression. If the critical maternal Chromosome-III element is the NPF gene locus, then $F_1$ offspring with maternal deletions of this chromosomal region may prevent inheritance of ethanol preference, much like not having inherited any maternal copies of Chromosome-III (*Figure 5d*). Using females with one Chromosome-III carrying a large deletion of the NPF gene region and one copy of wild-type NPF on a balancer Chromosome-III allowed us to ask whether an intact maternal NPF gene region was necessary for $F_1$ inheritance of ethanol preference. We found that legacy $F_1$ flies from unexposed mothers had no preference for ethanol, regardless of whether they inherited an intact NPF gene on a balancer chromosome or a chromosomal deletion of the NPF region (*Figure 5e–f*). Legacy $F_1$ flies from exposed mothers inheriting a wild-type NPF on a balancer chromosome exhibited a strong preference for ethanol, suggesting that multiple rearrangements, deletions and mutations of a balancer Chromosome-III are not sufficient to prevent ethanol preference in $F_1$ flies. By contrast, legacy $F_1$ flies from exposed mothers inheriting a Chromosome-III deletion of the NPF gene region do not inherit any preference for ethanol (*Figure 5f*). This was true for two different Chromosome-III deletions at the NPF locus, whereas a Chromosome-III deletion that does not disrupt the NPF gene had no effect (*Figure 5f*). Paternally inherited Chromosome-III deletions were not sufficient to prevent ethanol preference in $F_1$ flies (*Figure 5f*).

## Materials and methods

### Fly husbandry

Flies were maintained at room temperature on standard cornmeal-molasses media. A list of fly lines and genotypes used is reported in *Supplementary file 5*. Female flies were considered mature adults at three to five days post eclosion. Flies outside of this age range were not used for experimentation unless specifically noted, as for example in *Figure 1—figure supplement 1d*. Experiments involving manipulation of the maternal genotype, such as the maternal NPF knockdown, had a crossing scheme to avoid transgene expression in the $F_1$ generation. Virgin females with the

genotype of interest were crossed to *y,w* males and offspring were scored by eye color to ensure that flies assayed were not carrying both the Gal4 and UAS constructs.

## Wasp husbandry

The Figitid larval endoparasitoid *Leptopilina heterotoma* (strain Lh14) was used. In order to propagate wasp stocks, we used adult *D. virilis* in batches of 40 females and 15 males per each vial (Genesse catalog number 32–116). Adult *D. virilis* were allowed to lay eggs in standard Drosophila vials containing 5 mL standard Drosophila media supplemented with live yeast (approximately 25 granules) for one week before being replaced by adult wasps, using 15 female and six male wasps, for infections. Prior to wasp addition, vials were supplemented with approximately 500 µL of a 50% honey/water solution applied to the inside of the cotton vial plugs. Organic honey was used as a supplement. Wasps aged 3–7 days post eclosion were used for all infections and experiments, and were never reused for experiments.

## Wasp-exposure

Mature adult flies were used for wasp exposures: 40 female flies, 10 male flies, and 20 female Lh14 (*Leptopilina heterotoma*) wasps were placed in a vial with cornmeal-molasses media. This cohabitation (wasp exposure period) lasted for four days. The unexposed control consisted of the 40 female flies and 10 male flies with no wasp cohabitation. Both treatment groups were maintained at room temperature (approximately 22° C) with a 12 hr light-dark cycle for the duration of the exposure period.

At the conclusion of the exposure period, flies were separated into two cohorts. Following the removal of all wasps, one group of flies was used to propagate the next generation, while the second group was assayed for ethanol preference. Group one was placed on molasses-based embryo collection plates, supplemented with yeast paste, for egg collection. The collection period lasted for 24 hr, at which point the adult flies were removed. First instar larvae were transferred from these embryo plates to standard media vials. Larvae were density controlled to approximately 40 larvae per vial.

The second group was assayed for ethanol preference using a food-choice assay (*Kacsoh et al., 2015*). Briefly, five female flies and one male fly were placed into a modified petri dish with mesh top, termed the 'fly corral'. Two food sources were placed at opposite ends of the 'fly corral'. Each food source consisted of 0.45 g of instant drosophila media, hydrated with 2 mL liquid. Control food was hydrated entirely with distilled water, where as ethanol food was prepared with distilled water and a final addition of 95% ethanol to the top of the prepared food, creating a food with 6% ethanol by volume. Food sources were removed and replaced after 24 hr. Figures report the egg laying behavior of the first 24 hr interval unless otherwise noted. Total number of eggs laid on each food source was counted in a blinded fashion with treatment unknown to the counter. These egg counts are reported as a proportion of eggs laid on ethanol food. Flies that encountered ethanol-containing food were excluded from additional experimentation or lineage propagation. Fly corral experiments had ten replicates (cages) per condition.

## Transgenerational behavior experiments

Legacy flies, those descending from either the unexposed or exposed treatment, were divided into cohorts as described above for behavioral assay or embryo collection. These flies were not re-exposed to wasps except in the instance of multigenerational exposure experiments. Two experiments were conducted that involved multiple generations of treatment. For the successive exposures, three groups of flies were assayed; exposed legacy (two generations), exposed legacy (one generation), and unexposed legacy. In this instance, the exposed legacy (two generations) group was generated by subjecting $F_1$ exposed legacy flies to an additional round of wasp exposures. These flies therefore had grandparental and parental wasp exposure. Exposed legacy (one generation) had parental wasp exposure only (*Figure 1—figure supplement 1B*). It is important to note that the parents of the 'exposed legacy (one generation)' flies were $F_1$ unexposed legacy flies, and therefore had the same density control and egg collection as the other groups for the multigenerational duration of the experiment.

It is critical to note that baseline ethanol preference is highly variable depending on environmental conditions. Key factors are temperature and humidity, all ethanol oviposition assays were conducted in an environmentally controlled room at 25°C, approximately 30% humidity (±10%) with overhead lighting and a 12 hr light/dark cycle. Despite these controls, baseline ethanol preference varies day-to-day. For this reason, all groups for direct comparison (used in statistical tests) were tested at the same time.

Pertaining to the nonconsecutive exposure experiments; again three groups were assayed, the exposed legacy $F_8$ (two generations), exposed legacy (one generation), and the unexposed legacy. For these experiments, the exposed legacy $F_8$ (two generations) group was created by subjecting $F_7$-exposed legacy flies to an additional round of wasp exposures. These flies had a six-generation gap between ancestral wasp exposures. Flies in the exposed legacy (one generation) group were produced by exposing $F_7$ unexposed legacy flies to wasps, and collecting the subsequent offspring.

Several experiment specific modifications were made to the methods described above. To parse the maternal and paternal contributions to the inheritance of ethanol preference, two experiments were conducted. First, 40 mated female flies were used for wasp exposure, in the absence of males. Ten males were added to the population for the embryo collection period. For paternal contribution, male flies were removed from the exposure chamber and mated to unexposed virgin females. To test the role of vision in maternal inheritance, blind female flies mutant in *ninaB*[1], were crossed to wild type (CS) males. The reciprocal experiment crossed *ninaB*[1] males to CS female. These experiments were run in parallel and wasp exposures were preformed as previously described.

Compound chromosome experiments crossed two fusion stocks together (either chromosome-II or chromosome-III). The fusion lines retained phenotypic markers, and offspring with maternal or paternal chromosomes were sorted accordingly. Deficiency lines were crossed to CS flies and the genotype of the offspring (balancer or deficiency) was inferred from phenotypic markers.

Particular modifications for the $Orb2^{\Delta Q}$ memory-mutant experiments included an extra day of embryo collection. Following three-days of wasp exposure, flies and wasps were moved to the embryo collection chamber for the final treatment day. Eggs were collected for 24 hr in the presence of the 20 female Lh14 wasps. At the end of this period, wasps were removed and a new embryo collection plate was introduced for the second day of embryo collections. This second day of collection corresponds to the standard embryo collection timeframe in the above-described experiments. $F_1$ flies had the same genotype as the parental line.

Conditional knockdown of Orb2 was performed as previously described (*Bozler et al., 2017*). Briefly, flies were bred to have the following transgenes: mushroom-body-Gal4(switch) >UAS-Orb2 [RNAi]. The Gal4(switch) driver line contains a Gal4 transcription factor that is active only in the presence of the drug RU486. Vials were prepared with instant food containing two grams of instant drosophila food, hydrated with 8 mL of either RU486 (0.22 mg/mL) in 5% methanol, or vehicle only (5% methanol). Flies and wasps were cohabitated in the prepared vials and were transferred to new food each day regardless of treatment for three days. On the fourth day, an embryo collection was performed on instant food supplemented with yeast ($F_1$ collection during wasp exposure). The fifth day the wasps were removed and a second egg lay was conducted ($F_1$ collection post wasp exposure). F1 legacy flies were transferred and raised on cornmeal media, without the addition of vehicle or RU486 and therefore should have fully intact mushroom body and long-term memory capacity.

Sibling cohorts were collected to assess the longevity of the germline change. 'Brood 1' flies were collected in the 24 hr immediately following the removal of the wasps. 'Brood 2' flies were collected from the same parents, 10 days after the termination of the wasp exposure.

Finally, diet restriction experiments had two groups: one with high protein and the other low protein diets. Low protein flies were maintained on molasses based embryo plates. The high protein group was maintained in similar fashion, but with the addition of yeast paste. High/low diet was maintained for four days prior to embryo collection.

## Apoptosis quantification

Following the treatment period, ovaries were dissected and fixed in 4% formaldehyde for 30 min. Samples were stained with DAPI and apoptosis was scored based on the morphology of the nurse cell DNA. A researcher blinded to the genotype and treatment group of the samples performed the scoring. At a minimum, 15 ovaries were scored across three replicates (independent wasp exposures) for each group.

## Immunostaining and microscopy

Antibody to neuropeptide F was generated in a rabbit to the full length NPF peptide: C-Ahx-SNSRPPRKNDVNTMADAYKFLQDLDTYYGDRARVRFamide. The antibody was subsequently purified using a truncated peptide containing the first 28 amino acids of NPF. Following purification, the antibody was depleted using a peptide of the eight amino acid C-terminal tail, shared by many neuropeptides. All peptide synthesis, antigen injection, serum preparation, peptide purification, and depletions were performed by 21st Century Biochemicals.

Whole flies were fixed in 4% formaldehyde overnight at 4° C. Female brains were dissected, blocked, and incubated with anti-NPF (1:1000) overnight at 4° C. Antibody solution was removed and samples were blocked before the addition of the secondary antibody, anti-rabbit 488 (1:200), at room temperature for two-hours. Samples were counter stained with DAPI.

For NPF quantification, flies expressing a RFP tagged histone were dissected along with treatment groups and stained in the same solution. Pixel intensity of the fan shaped body (FSB) was measured in Image J. The FSB was outlined by hand and intensity measured. A background measure was made of the region immediately ventral to the FSB, with the same total area as the outlined FSB. The background value was subtracted from FSB measurement. Finally, the background-adjusted intensity value for each brain was divided by the arc length of its' FSB. This process was repeated for each treatment group and the corresponding histone-RFP flies. These values were normalized to the histone-RFP flies to serve as a control for batch specific variation in staining. Each treatment group was normalized to the unexposed average of that replicate using the formula(s):

$$\text{Flurescence} = (\text{FSB}_{\text{intensity}} - \text{background}_{\text{intensity}})/\text{FSB}_{\text{length}}$$

$$\text{BatchNormalized} = (\text{Fluorescence}_{\text{CantonS}}/\text{Fluorescence}[\text{avg}]_{\text{his-RFP}})$$

$$\text{AFU} = \text{BatchNormalized}_{\text{exposed}}/\text{BatchNormalized}[\text{avg}]_{\text{unexposed}}$$

Standard fluorescent images were visualized with the Nikon Eclipse E800 microscope and a Olympus DP71 camera. For each experiment, wasp exposure and staining were performed on two separate occasions and final data was pooled after checking for the absence of a batch effect. A minimum of 10 brains were dissected for each treatment replicate as well as RFP-histone co-staining brains. Final quantified sample size range from 15 to 20 (normalized brains), due to sample loss or damage. Imaged samples were only excluded if clear damage or trauma (from dissection or staining process) was evident in the region of interest (FSB or P1 nuerons).

## RNA quantification

Mature female flies were anesthetized with $CO_2$ and collected in 15 mL conical tubes, either immediately following the treatment period ($F_0$), or 3–5 days post eclosion ($F_1$-$F_2$). Flies were frozen in liquid nitrogen and briefly vortexed to separate whole heads. Approximately 100 heads were collected for each replicate. A miRNeasy Kit (Qiagen) with on-column DNase treatment was used for RNA isolation. Four samples of each treatment group were prepared.

RNA samples were depleted of rRNA followed by random priming. Minimum sequencing depth per sample was 40 million paired-end reads on the Illumina platform. Sequencing reads were indexed to transcripts using Kallisto and the Ensembl genome (BDGP6) with 100 bootstraps (*Aken et al., 2017*; *Bray et al., 2016*). Downstream processing and statistical analyses used Sleuth (*Pimentel et al., 2017*). Heat maps were generated using hierarchical clustering and the R package pheatmap.

NPF transcript was measured by qPCR (SYBR Green, Thermo-Fisher 4309155). NPF primer targeted mRNA (TCCTGGTTGCCTGTGTGG, TCAGCCATAGTGTTGACATCG). Actin served as the control gene (CGCAAGGATCTGTATGCCAA, ACGGAGTACTTGCGCTCTGG). Fold change was calculated using the delta-delta Ct method.

## Statistics

Statistical tests were run in R (3.0.2 version, 'Frisbee Sailing'). P-values for egg count data, NPF staining, and apoptosis quantification, were produced by applying a Mann-Whitney Rank Sum test. Error

bars presented in the egg count ethanol preference graphs are bootstrap confidence intervals, generated using the boot package.

## Discussion

Perhaps the blank slate has more written on it than we once thought. Indeed, it would appear that animals are bound to their ancestors in a way that some might consider Lamarckian (*Galloway and Etterson, 2007*; *Herman and Sultan, 2011*; *J. Marshall and Uller, 2007*). The ethanol preference we observed in this study is heritable but modifiable and responsive to environmental cues, as it can be enhanced or decay across generations. Our data suggest that there is an ultimate return to pre-wasp exposed state by the $F_6$ generation. If there are lingering effects of wasp exposure beyond this generation, they are not detected in our assays. Not only does the ethanol preference behavior revert to unexposed levels, but we also detected no priming or enhancement effect in the $F_8$ generation following a second wasp exposure (*Figure 1—figure supplement 1f*).

Inheritance of ethanol preference requires several factors: We found that the initiation of the epigenetic program in the founding generation ($F_0$) is maternal in nature, and requires effector caspases in the female germline. However, continuation of the epigenetic program throughout the remaining generations is distinctly different in several ways. Both male and female progeny ($F_1$) are able to pass on ethanol preference to their offspring. Although, it is possible that the $F_1$ generation requires germline effector caspases for the transmission of the ethanol preference, the lack of female germline apoptosis and paternal ability to confer this behavior points to a caspase-independent maintenance mechanism. A further, and curious, distinction between the generations is in the ethanol preference itself, as it persists in the $F_1$ generation, rather than mirroring the $F_0$ generation and decaying over 10 days.

The unifying mechanism behind many of these observations is the central role of NPF signaling in this system. Governing both germline apoptosis and the ethanol preference neuronal NPF signaling modulates the ethanol preference as well as its inheritance. Maternal imprinting of the NPF locus or nearby regions has a dominant effect, leading to the possibility that the $F_1$ paternal locus is imprinted in trans. It is tempting to speculate on the role of canonical imprinting mechanisms, such as the Polycomb repressive complexes, although a molecular apparatus remains elusive for the time being (*Figure 6*).

This multi-generational ethanol preference underscores the importance of environmental conditions on behavior and physiology. Numerous studies have indicated that we may need to look beyond the individual, to longer lasting and persistent effects of environmental stresses. This study illustrates the complexity of inheritance and highlights the incredible resiliency and plasticity of organisms to adapt to changing circumstance. Of particular interest is the conserved functions of NPF and its mammalian homolog NPY in modulating a variety of human behaviors, including stress responses and alcohol abuse disorders (*Thorsell and Mathé, 2017*). Our studies raise the intriguing possibility that NPF/NPY and their receptors could be subject to epigenetically modified states determined by parental environment and experience. Germline inheritance of epigenetically modified neuro-signaling networks, such as those modulated by NPF/NPY, could be one mechanism through which transgenerational inheritance of behavioral predispositions persist, as reported here for Drosophila. It should be noted that such epigenetically inherited behaviors that persist for multiple generations could be interpreted as dominant familial genetic traits. If mammalian NPY is inherited in epigenetically modified states, then this would require a fundamental change in how we study and view inheritance of NPY-related behavioral disorders and possible effects of parental environment.

## Additional information

### Funding

| Funder | Grant reference number | Author |
| --- | --- | --- |
| National Institutes of Health | 1DP1MH110234 | Giovanni Bosco |
| Defense Advanced Research Projects Agency | HR0011-15-1-0002 | Giovanni Bosco |

| National Institutes of Health | T32-GM009704 | Julianna Bozler |
|---|---|---|

The funders had no role in study design, data collection and interpretation, or the decision to submit the work for publication.

## Author contributions

Julianna Bozler, Conceptualization, Formal analysis, Investigation, Writing—original draft; Balint Z Kacsoh, Investigation, Writing—review and editing; Giovanni Bosco, Conceptualization, Funding acquisition, Writing—review and editing

## Author ORCIDs

Julianna Bozler (iD) https://orcid.org/0000-0002-6275-3409
Balint Z Kacsoh (iD) https://orcid.org/0000-0001-9171-0611
Giovanni Bosco (iD) https://orcid.org/0000-0002-8889-9895

## Decision letter and Author response

Decision letter https://doi.org/10.7554/eLife.45391.026
Author response https://doi.org/10.7554/eLife.45391.027

## Additional files

### Supplementary files

• Supplementary file 1. Statistical tests and p-values relating to main text figures.
DOI: https://doi.org/10.7554/eLife.45391.015

• Supplementary file 2. Oregon R experimental data. Key experiments were replicated using the additional wild-type strain OreR. 'Corresponding Figure' indicates the experiment that was replicated: A listing of *Figure 1B* therefore indicates that the experimental conditions for *Figure 1B* were duplicated using OreR flies.
DOI: https://doi.org/10.7554/eLife.45391.016

• Supplementary file 3. Canton S day-2 data; mean(s) and p-value(s).
DOI: https://doi.org/10.7554/eLife.45391.017

• Supplementary file 4. RNA sequencing results from female fly heads across generations. The beta value (b) is approximately analogous to the natural log fold change of the transcript, and the q-value is the measure of significance. Transcript meeting the threshold criteria (q-value and beta) for any one generation was included in the table (corresponding to the head map of *Figure 1—figure supplement 4*).
DOI: https://doi.org/10.7554/eLife.45391.018

• Supplementary file 5. Drosophila stock list and source information.
DOI: https://doi.org/10.7554/eLife.45391.019

• Transparent reporting form
DOI: https://doi.org/10.7554/eLife.45391.020

### Data availability

RNA sequencing data is stored under the NCBI Sequence Read Archive, BioProject ID PRJNA414223.

The following dataset was generated:

| Author(s) | Year | Dataset title | Dataset URL | Database and Identifier |
|---|---|---|---|---|
| Bozler J, Kacsoh BZ, Bosco G | 2019 | Drosophila female head transcriptome | https://www.ncbi.nlm.nih.gov/bioproject/PRJNA414223/ | NCBI BioProject, PRJNA414223 |

The following previously published dataset was used:

| | | | | Database and |
|---|---|---|---|---|

| Author(s) | Year | Dataset title | Dataset URL | Identifier |
|---|---|---|---|---|
| Bozler J, Kacsoh BZ, Bosco G | 2018 | Drosophila female head transcriptome | https://www.ncbi.nlm.nih.gov/bioproject/PRJNA414223/ | NCBI BioProject, PRJNA414223 |

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
