## [Decision Letter]

Thank you for submitting your article "Transgeneratonal inheritance of ethanol preference is caused by maternal NPF repression" for consideration by *eLife*. Your article has been reviewed by three peer reviewers, including Leslie C Griffith as the Reviewing Editor and Reviewer #1, and the evaluation has been overseen by Maarten van Lohuizen as the Senior Editor. The following individual involved in review of your submission has agreed to reveal their identity: Ralph Greenspan (Reviewer #2).

The reviewers have discussed the reviews with one another and the Reviewing Editor has drafted this decision to help you prepare a revised submission.

Please note that *eLife* offers supplements to specific figures, rather than a supplemental section. If you look at *eLife* papers you will see how powerful it is to use figure supplements for additional information relevant to that specific figure. We also have no limits on the number of figures, so feel free to add additional figures to the Materials and methods section if that would be helpful. In any event, please try to bring your paper into conformity with *eLife*'s format.

Summary:

In this study, the authors investigate the mechanism of transgenerational inheritance of egg-laying preference in ethanol. They illustrate the disparate ways (experience and development) of exhibiting the preference, and show how the same experience-acquired information (preference) can be manifest through different substrates (neural, germline). The most important findings are that mechanistic requirements of preference alteration within a generation are partially overlapping, yet distinct from the mechanistic requirements of preference inheritance across a generation; they also demonstrate the requirement of a maternal 3rd chromosome in inheritance across generation, implicating imprinting of the NPF gene.

Essential revisions:

1) An important distinction between preference in exposed flies vs. their progeny is the requirement of memory. The experiment in Figure 1C suggests some degree of dissociation between long term memory of preference and transgenerational inheritance of preference, as long-term memory impairment partially but not completely impairs the transgenerational inheritance. Is this partial transgenerational deficit because the parent can't hang onto the memory, a situation akin to "brood 2" (Figure 1—figure supplement 2) where the parental memory is gone, or does the lack of Orb2 in the mutant impair the inheritance of the offspring (not an unreasonable idea given the role of Orb2 in development)? To distinguish these two models, the authors should do one of several things: impair memory by expressing Orb2 RNAi in the adult mushroom body and assessing the inheritance, or use another memory mutant, or use some other acute MB manipulation.

2) The authors need to clean the paper up and provide the necessary info in figure legends for the reader to understand what they have done.

Specific points:

1) In the Introduction, “wasp” should be “wasps”.

2) There are many grammatical errors throughout the paper e.g. “treated to” ethanol.

3) Do flies exposed to wasps prefer to *consume* EtOH as well? Are they more likely to be addicted? Just curious.

4) Figure 1C is not completely clear. The flies assayed are ALL F_1_? Perhaps label this better?

5) Are the dcp1 and drice knockdowns in ovary only? What is the driver? This information should be added to the legend and text.

6) Figure 3.

A)Transgene controls (UAS-NPF and UAS-NPRRNAi) are missing.

B) There is no label for what the bars are in Figure 3B/C.

C) Why are only some of the pairwise comparisons shown? Are all others N.S.? If so, this should be stated.

D) It’s not completely clear in the text if the legacy RNAi flies also have both the driver and RNAi? I would guess not (were they mated to WT males? other males?) but you should state this explicitly in the Results.

E) Where is NPFR knocked down? Brain? Ovary? Is there NPF in ovary/reproductive tract?

F) Where is NPF overexpressed? Brain? Ovary?

7) In general, the figure legends are not well written. They lack a huge amount of detail that the reader needs to evaluate the data, e.g. N, the type of statistical tests used, labels, genotypes etc.

8) Figure 4. What was the ROI for the quantification? Cell body? Neuropil? Maybe show on the image?

9) Figure 5. Blind? How? It actually matters – many 'visual' mutations are pleiotropic. A genotype/condition must be given in either text or legend.

10) Results, subsection “Maternal Chromosomal Inheritance of Ethanol Preference Behavior”: there actually are paternally-inherited regulator RNAs, at least in mice, so this is not completely clear cut.

11) Figure 3E heading says overexpression and knockdown but the data shows only knock down.

12) Transcriptomic data: clarify whether changes are relative to unexposed flies of the same generations or relative to exposed flies of different generations. For example, are the F_1_ and F_2_ changes relative to F_0_ exposed flies or F_1_/F_2_ unexposed flies? At what point do transcriptomes change?

13) Figure 1—figure supplement 3A suggests female genotype is not important for transgenerational inheritance, while Figure 5 deals with the necessity of female genotype. This needs better clarification.

There are also a few things that the authors might consider adding/discussing as they would enhance the story; these are not required revisions.

1) A supplementary figure describing the compound autosome technique for chromosomal localization would be helpful, given the general lack of familiarity with this genetic technique of the vast majority of researchers.

2) NPF is involved in two very distinct events: acute and relatively unstable preference for ethanol in F_0_ and relatively long-lived and stable preference for 5 generations. It is unclear how the NPF signals are regulated differently to accommodate these disparate time scales, considering there is no change in mRNA level in F_0_ and F_1_, although protein level is reduced. One plausible explanation would be tissue-specific transcriptional repression. It would be important to do RNA in situ for NPF both in the F_0_ and F_1_ head, particularly focusing on fan shaped body and P1 neurons. Also, is NPF expressed in the germ cells, and what is happening at the NPF genomic locus in germline cells of exposed females?

3) What is the significance of the difference in NPF levels in P1 neurons in F_0_ vs F_1_? Would increase in NPF in P1 abolish experience-dependent changes in preference for F_0_ while leaving all its progenies' preferences intact?

4) What happens to ethanol preference memory when NPF is deleted on maternal chromosome. These flies can't inherit preference-but can they learn it themselves? Also. what happens to their fan body and P1 NPF levels?

5) How specific are transgenerational effects generally? For example, does starvation-mediated apoptosis lead to transgenerational alteration in starvation responses? The fact that starvation leads to apoptosis but not ethanol preference is interesting in light of this theme.

6) What mediates the "decay" of this memory in varying time scales?

---

## [Author Response]

Essential revisions:1) An important distinction between preference in exposed flies vs. their progeny is the requirement of memory. The experiment in Figure 1C suggests some degree of dissociation between long term memory of preference and transgenerational inheritance of preference, as long-term memory impairment partially but not completely impairs the transgenerational inheritance. Is this partial transgenerational deficit because the parent can't hang onto the memory, a situation akin to "brood 2" (Figure 1—figure supplement 2) where the parental memory is gone, or does the lack of Orb2 in the mutant impair the inheritance of the offspring (not an unreasonable idea given the role of Orb2 in development)? To distinguish these two models, the authors should do one of several things: impair memory by expressing Orb2 RNAi in the adult mushroom body and assessing the inheritance, or use another memory mutant, or use some other acute MB manipulation.

The reviewers’ point is well taken. To address this possibility we have added two additional experiments (Figure 1—figure supplement 1). First, the experiment was repeated with the memory mutant *amn*^1^, thereby addressing the concerns about an Orb2 specific effect. Secondly, we used a conditional knockdown of Orb2 in the mushroom body; thus generating F_1_ progeny that did not have expression of the RNAi transgene.

2) The authors need to clean the paper up and provide the necessary info in figure legends for the reader to understand what they have done.

We have made particular changes to address the specific comments.

Specific points:1) In the Introduction, “wasp” should be “wasps”.

We have corrected this typo in the text.

2) There are many grammatical errors throughout the paper e.g. “treated to” ethanol.

We apologize for the grammatical errors in the previous draft. We have taken additional steps to proof read and correct these errors.

3) Do flies exposed to wasps prefer to consume EtOH as well? Are they more likely to be addicted? Just curious.

The reviewer raises a very interesting question. Although we have previously shown that adult flies continue to eat following wasp exposure, we do not have any direct data to conclusively show that they preferentially consume ethanol. Unfortunately, to investigate this question one would need a very different experimental design and specialized apparatus. While this is an area of interest for us, we feel that it is better suited to a future investigation and outside of the scope of the current study.

4) Figure 1C is not completely clear. The flies assayed are ALL F_1_? Perhaps label this better?

We apologize for the confusion. In Figure 1C data is shown for legacy (F_1_) flies, collected during or post wasp exposure. We have clarified this in the figure itself as well as in the figure legend.

5) Are the dcp1 and drice knockdowns in ovary only? What is the driver? This information should be added to the legend and text.

The driver line used in these experiments is specific to the female germline. We have added the transgene information into the text in hopes of clarifying this point: “Nevertheless, maternal germline knockdown of the effector caspases Dcp-1 and drice (maternal-αtubulin-Gal4 > UAS-Dcp-1[RNAi], and maternal-αtubulin-Gal4 > UAS-Drice[RNAi] respectively) produce offspring without an ethanol preference, regardless of parental treatment (Figure 2C).”

We have also added this information into the figure legend: “Genetic knockdown of the germline effector caspases, Drice or Dcp-1, was achieved by expressing a RNA hairpin in the female germline (driven by the maternal-αtubulinGal4).”

6) Figure 3.A)Transgene controls (UAS-NPF and UAS-NPRRNAi) are missing.

We have added in these data as a supplemental figure (Figure 3—figure supplement 1). We did not include them in the primary graph because the experiments were conducted at different times and we felt that this might invite inappropriate statistical comparisons. Importantly, both transgene lines behaved similarly to wild type.

B) There is no label for what the bars are in Figure 3B/C.

We apologize for the oversight; we have added labeling for the violin plots in Figure 3.

C) Why are only some of the pairwise comparisons shown? Are all others N.S.? If so, this should be stated.

The results of all of our statistical tests are shown in the figure. Statistical tests that yielded a non-significant result are indicated with “N.S” in the figure itself. The chosen statistical comparisons were selected to test specific hypotheses: First, comparisons between treatment group of the same genotype were performed to assay for changes due to wasp exposure. Secondly, we explored the role of NPF signaling, manipulated through genetic means without wasp exposure. Therefore, the unexposed NPF-manipulation group was compared to the unexposed background line. We restricted our statistical tests so that they were hypothesis driven, and although significance for all pairwise comparisons could be calculated we feel that all pairwise comparisons would not be informative and may be statistically inappropriate. For instance, conducting all pairwise tests would necessitate the comparison between samples differing both in treatment and genotype, resulting in a meaningless significance value (in other words, comparisons that are not considering the carefully paired control group for each experimental treatment and/or genotype).

D) It’s not completely clear in the text if the legacy RNAi flies also have both the driver and RNAi? I would guess not (were they mated to WT males? other males?) but you should state this explicitly in the Results.

The reviewers are correct, the legacy RNAi flies are not carrying both transgenes, as this would have greatly complicated the interpretation of our results. We have stated this more clearly in the Results section: “For these experiments, it was critical to ensure that F_1_ flies did not share the maternal genotype, and a crossing scheme was devised to avoid progeny with transgene expression (see Materials and methods section).”

E) Where is NPFR knocked down? Brain? Ovary? Is there NPF in ovary/reproductive tract?

The driver line used in the NPF-R experiments was a pan-neuronal driver (ElavGal4). We have added this to the figure legend: “Knockdown of the NPF-receptor in neurons (using the pan neuronal driver Elav-Gal4) leads to increased germline apoptosis”.

F) Where is NPF overexpressed? Brain? Ovary?

The genetic NPF experiments (overexpression and knockdown) used the NPF-Gal4 line. We have added this detail to the figure legend: “NPF overexpression (OE) or knockdown (KD) was achieved in the NPF-expression pattern (NPF-Gal4).”

7) In general, the figure legends are not well written. They lack a huge amount of detail that the reader needs to evaluate the data, e.g. N, the type of statistical tests used, labels, genotypes etc.

We have reworked the figure legends to include greater experimental and statistical detail. We have added information regarding statistical measures, genotype, and sample size (sample size for the ethanol preference was not added, as the n of 10 was standard across experiments).

8) Figure 4. What was the ROI for the quantification? Cell body? Neuropil? Maybe show on the image?

We thank the reviewers for this suggestion. We have generated a supplemental figure that addresses the regions of quantification (Figure 4—figure supplement 2).

9) Figure 5. Blind? How? It actually matters – many 'visual' mutations are pleiotropic. A genotype/condition must be given in either text or legend.

The flies used were *ninaB^1^* mutants. We have added clarification both in the figure itself as well as the figure legend.

10) Results, subsection “Maternal Chromosomal Inheritance of Ethanol Preference Behavior”: there actually are paternally-inherited regulator RNAs, at least in mice, so this is not completely clear cut.

We absolutely agree with the reviewers, and we did not mean to suggest that cytoplasmic factors (or small RNAs) can only be passed on maternally. Rather, we were trying to disentangle the ‘chromatin mark’ mechanism from the ‘cytoplasmic’ mechanism; which could be either maternal or paternal, however in this case we have shown that males do not participate in the initial transmission of the epigenetic program. We have removed the reference to small RNAs in an attempt to clarify our thinking in the text.

11) Figure 3E heading says overexpression and knockdown but the data shows only knock down.

We thank the reviewers for finding this error; the figure has been corrected.

12) Transcriptomic data: clarify whether changes are relative to unexposed flies of the same generations or relative to exposed flies of different generations. For example, are the F_1_ and F_2_ changes relative to F_0_ exposed flies or F_1_/F_2_ unexposed flies? At what point do transcriptomes change?

Each generation had a paired exposed and unexposed group. The fold change data was normalized within the generation (F_1_ exposed legacy normalized to F_1_ unexposed legacy). We have added text to the Results section to make this clearer: “Heads from the exposed legacy F_1_ and F_2_ generation were collected and normalized to their respective unexposed legacy group.”

13) Figure 1—figure supplement 3A suggests female genotype is not important for transgenerational inheritance, while Figure 5 deals with the necessity of female genotype. This needs better clarification.

Figure 1—figure supplement 3A deals specifically with the F_1_ males being able to pass on the ethanol preference to the F_2_ generation. Alternatively, Figure 5 explores the initial transmission from the F_0_ generation (exposed group) to the legacy flies. We believe these data point to differences between the initial transmission (following wasp exposure) and the transmission of the behavior between legacy flies (i.e. activation vs. maintenance). We have attempted to clarify this in several ways. First, we have more clearly labeled the graphs in Figure 1—figure supplement 3A. We have also added text to the Results section to better highlight this difference: “Interestingly, F_1_ offspring from an exclusively maternal wasp exposure inherit ethanol preference, while F_1_ offspring from an exclusively paternal exposure did not (Figure 5A). Importantly, this demonstrates a key difference between the F_0_ males and the F_1_ males, and may point to an ‘activation phase’ and ‘maintenance phase’ of the epigenetic program.”